# Accurate Conversion of Land Surface Reflectance for Drone-Based Multispectral Remote Sensing Images Using a Solar Radiation Component Separation Approach

**DOI:** 10.3390/s25082604

**Published:** 2025-04-20

**Authors:** Huasheng Sun, Lei Guo, Yuan Zhang

**Affiliations:** 1Shandong Provincial Key Laboratory of Soil and Water Conservation and Environmental Protection, School of Resources and Environment, Linyi University, Linyi 276000, China; sunhuasheng@lyu.edu.cn (H.S.); guolei1@lyu.edu.cn (L.G.); 2School of Geographic Sciences, East China Normal University, Shanghai 200241, China

**Keywords:** drone-based remote sensing, multispectral images, land surface reflectance, direct and scattering radiation, solar radiation component separation

## Abstract

Land surface reflectance is a basic physical parameter in many quantitative remote sensing models. However, the existing reflectance conversion techniques for drone-based (or UAV-based) remote sensing need further improvement and optimization due to either cumbersome operational procedures or inaccurate results. To tackle this problem, this study proposes a novel method to mathematically implement the separation of direct and scattering radiation using a self-developed multi-angle light intensity device. The verification results from practical experiments demonstrate that the proposed method has strong adaptability, as it can obtain accurate surface reflectance even under complicated conditions where both illumination intensity and component change simultaneously. Among the six selected typical land cover types (i.e., lake water, slab stone, shrub, green grass, red grass, and dry grass), green grass has the highest error among the five multispectral bands with a mean absolute error (MAE) of 1.59%. For all land cover types, the highest MAE of 1.01% is found in the red band. The above validation results indicate that the proposed land surface reflectance conversion method has considerably high accuracy. Therefore, the study results may provide valuable references for quantitative remote sensing applications of drone-based multispectral data, as well as the design of future multispectral drones.

## 1. Introduction

Drone-based remote sensing technology plays an important role in small-area remote sensing monitoring due to many prominent advantages such as flexibility and maneuverability, high operation efficiency, low-cost data acquisition, high spatial and temporal resolution, and cloud cover avoidance. It has become an essential and irreplaceable complement to traditional satellite remote sensing. At present, drone-based remote sensing technology has been widely used in many fields, such as resource investigation, environmental monitoring, photogrammetry, 3D modeling, on-site monitoring and assessment of disasters, construction engineering, and urban planning. In recent years, technological developments and innovations have made both drones and sensors gradually progress toward diversification and miniaturization. The available sensors that can be mounted on drones include RGB cameras, multispectral cameras, hyperspectral cameras, thermal infrared cameras, synthetic aperture radars (SARs), and light detection and ranging (LIDAR) instruments.

Because of the crucial role of multispectral remote sensing images in quantitative remote sensing, this study focused on the data processing and quantitative applications of drone-based multispectral images. At present, there are several multispectral cameras that can be carried by drones, such as Parrot Sequoia and MICASENSE RedEdge-MX (note: they need to be mounted on drones like eBee, DJI M300 RTK, etc.), as well as some integrated drone-based multispectral imaging systems, such as DJI Phantom 4M and DJI Mavic 3M. By virtue of the low flight altitude of drones and the high-resolution cameras mounted on the drones, centimeter-level spectral features of ground objects can be obtained by the drone-based remote sensing imaging system. The images collected by multispectral cameras can serve various quantitative applications, such as calculating vegetation indices [1,2,3,4,5], leaf area index [6,7], and biomass/yield [8,9,10]; monitoring plant water/nutrient content [4,6,11,12,13,14,15,16], pests, and disease [17]; building remote sensing evapotranspiration models [18]; and monitoring soil carbon content [19,20,21], soil salt content [22], forest status [23,24,25,26,27], water quality [28,29], etc. Land surface reflectance is a basic physical parameter to describe the spectral features of ground objects and a vital input parameter in remote sensing models. In the above-mentioned quantitative applications, it is necessary to convert the digital numbers (DNs) of image pixels into accurate land surface reflectance.

Under stable illumination conditions, after the elimination of vignette effect and lens distortions, if an image including a group of (two or more) reference boards with different reflectance is taken during the photogrammetry process, the DNs recorded in each band can be directly converted into land surface reflectance through linear stretching calculations [30,31,32]. However, the above method not only has to rely on calibration boards but also usually generates inaccurate results [33,34,35] because of the extraordinarily complicated conditions, which are particularly evident in the following aspects: (1) The illumination conditions might change with varying cloud cover and solar altitude. (2) To capture high-quality images, the camera’s exposure time, aperture size, and ISO value need to be adjusted with the illumination conditions, and this might lead to inconsistency in the recorded DNs even for the same target under the same illumination conditions. (3) In some special sites such as dense forests, water surfaces, and swamps, it is difficult or impossible to place reference boards, and moreover, it is unrealistic to capture the reference boards in each image.

In order to be independent of the reference boards and simplify data processing, the empirical line method was used to obtain approximate results [30,36,37]. Additionally, many studies suggest installing a light intensity sensor on the drone to simultaneously record the illumination status during the image capture time, and the recorded results are then used to calculate land surface reflectance [31,32,33,35,38,39]. However, obtaining reliable ground-received irradiance is essential for achieving accurate surface reflectance. For an unstable platform, the measured result of light intensity sensor is inconsistent with the solar radiation intensity reaching the ground due to the changing altitude of aircraft during flight. As a result, the calculated land surface reflectance is not accurate. To address the issue, some studies use the block correction method to reduce the tilting effect [38,39,40]. However, the obtained results are usually approximate rather than accurate, because the light conditions might change during the image capture time due to changes in weather and solar altitude. Additionally, some studies employed the cosine function to correct the directly measured results, but experimental tests indicated that the cosine correction is still inadequate [31,34] because solar radiation includes not only direct radiation but also scattering radiation. On the one hand, the composition of solar radiation varies greatly under different weather conditions, e.g., direct radiation is dominant (>85%) on clear days, and it becomes weak as the cloudiness increases, even when there is no direct radiation at all on overcast days; on the other hand, the tilting effect on direct radiation and scattering radiation is totally different. To solve this problem, Sun et al. [41] proposed an effective method to correct the directly measured results and yield more accurate results under the assumption that the direct radiation proportion remains stable over a short period of time. However, the above assumption is obviously idealistic, so the method still cannot adapt to complicated illumination conditions with dramatic fluctuations in solar radiation components on cloudy days.

Furthermore, in order to obtain more accurate solar irradiance results, the light intensity sensor can be installed on an automatic leveling device. For example, Markelin et al. [32] used a balance system to compensate the tilting effect of the light intensity sensor within 15°, but for a multi-rotor drone, sometimes the tilting angle can exceed 30°. If a precise leveling device (e.g., a 3D stabilization gimbal) is installed in the light intensity sensor, it will greatly increase the burden and manufacturing cost of the aircraft.

In practical applications, to obtain more reliable results from the irradiance sensor without a precise leveling device, the commonly used methods are listed as follows:It is recommended that the solar elevation angle is as large as possible (e.g., to capture images at noon), and the flight routes are perpendicular to the solar azimuth angle to minimize the impact of tilting effect; however, the method cannot completely eliminate the tilting effect.If a multi-rotor drone is used as the platform, hovering mode is recommended to minimize the impact of tilting effect. However, the light intensity sensor can only remain horizontal when there is no wind at all; otherwise, it cannot guarantee that the sensor is horizontal, and therefore, it still cannot eliminate the impact of tilting effect. Experimental tests have indicated that the tilting angle can exceed 15° in hovering mode when the wind speed is relatively higher. The primary advantage of hovering mode is that it can eliminate motion blurring, yet a significant drawback is that the operation efficiency is extremely low.

In summary, the existing land surface reflectance conversion methods for drone-based multispectral images are only suitable for simple and specific conditions. Until now, it remains challenging to directly convert drone-based images into accurate land surface reflectance data under complicated illumination conditions. In this study, a novel method was proposed to simplify the data processing process and improve data processing efficiency. Direct and scattering radiation can be separated through mathematical methods based on the tilting measurement results facing different directions, and the above operation plays a crucial role in this study. Based on the separated results, accurate land surface reflectance data are expected to be obtained directly even under complicated illumination conditions.

## 2. Theory and Methods

### 2.1. Solar Radiation Component Separation

Solar radiation is the direct source for drone-based remote sensing. Because of the atmosphere surrounding the Earth, the incident radiation on the ground includes not only direct radiation from the sun but also atmospheric scattering radiation from the sky. The radiation properties are entirely different, so separating direct and scattering radiation is crucial for converting drone-based multispectral images into land surface reflectance data.

In the field of meteorology, a pyranometer can be used to measure solar radiation. For direct solar radiation measurement, it requires a narrow field of view observation to block scattering radiation and exclusively receive the direct part; on the contrary, for scattering measurement, a shading ring is used to block the direct solar radiation and retain only the scattering part. However, the shading ring itself has some area, so it will block part of the scattering radiation. Even if scattering compensation is applied to the measured result, the corrected result is still inaccurate, because the shading ring is not an ideal black body and it has some reflection. In addition, for the scattering radiation sensor with a shading ring, it is necessary to adjust the shading ring to the corresponding position according to the solar declination so as to guarantee the shading ring always blocks the direct solar radiation during the moving process of the sun. Therefore, the operational process is rather cumbersome; moreover, the pyranometer is suitable only for ground observation, and it cannot be mounted on a moving aircraft for real-time measurement. Currently, there are no better methods for solar radiation component separation.

To solve the above-mentioned problems of the traditional technique, in this study, a new approach was proposed to achieve the separation of direct and scattering radiation by using several completely identical sensors facing different directions. Consequently, the method described in the paper of Sun et al. [41] can be used to correct the tilting effect of the light intensity sensor mounted on a drone, and then the drone-based multispectral images can be further converted into accurate surface reflectance data. The details are described below.

For a light intensity sensor, the solar radiation received includes direct solar radiation, atmospheric scattering, and a small part of ground reflection. Their quantitative description is as follows:(1)Direct solar radiation(1)$${E^{\prime }}_{dir}=E_{dir}cosz$$
where ${E^{\prime }}_{dir}$ is the direct irradiance received by a tilted sensor plane; $E_{dir}$ is the direct irradiance of a sensor perpendicular to the incident direction; and z is the incident angle of the sensor plane. It should be noted that if z≥90° (i.e., cos(z)≤0), the direct radiation cannot reach the front side of the sensor, and it means that the front side is in the shadow and no direct radiation is received, although it exists. In the above case, ${E^{\prime }}_{dir}=0$.
(2)Atmospheric scattering radiation

(2)$${E^{\prime }}_{sca}=E_{sca}cos^{2}(s/2)$$
where ${E^{\prime }}_{sca}$ is the scattering irradiance received by a tilted sensor; $E_{sca}$ is the scattering irradiance received by a horizontal sensor; and s is the slope angle of the sensor plane.
(3)Ground reflection radiation

(3)$${E^{\prime }}_{ref}=E_{ref}sin^{2}(s/2)={\bar{R}}_{g}E_{g}sin^{2}(s/2)$$
where ${E^{\prime }}_{ref}$ is the ground reflection irradiance received by a tilted sensor; $E_{ref}$ is the ground reflection irradiance; ${\bar{R}}_{g}$ is the average ground reflectance, and in visible and near-infrared spectral range, usually ${\bar{R}}_{g}=0.2$ for common ground and ${\bar{R}}_{g}=0.7$ for snow ground approximately. Note: For common ground, the ground reflection radiation received by a tilted surface is weak, e.g., s=30°, ${E^{\prime }}_{ref}=0.052E_{g}$, s=60°, ${E^{\prime }}_{ref}=0.1E_{g}$; s=90°, ${E^{\prime }}_{ref}=0.1414E_{g}$; and $E_{g}$ is the solar irradiance received by the horizontal ground (note: s=0°, so it just involves the direct solar radiation and the atmospheric scattering radiation), and its expression is shown in Equation (4).(4)$$E_{g}=E_{dir}cos\theta +E_{dif}$$
where θ is the incident angle of direct solar radiation.

The total solar irradiance (E) directly measured by a tilted (s≠0°) sensor plane involves the above-mentioned three parts, which can be described by Equation (5).(5)$$\begin{array}{l} E={E^{\prime }}_{dir}+{E^{\prime }}_{sca}+{E^{\prime }}_{ref}\\ \;=E_{dir}cosz+E_{sca}cos^{2}(s/2)+{\bar{R}}_{g}(E_{dir}cos\theta +E_{sca})sin^{2}(s/2)\\ \;=E_{dir}[cosz+{\bar{R}}_{g}cos\theta sin^{2}(s/2)]+E_{sca}[cos^{2}(s/2)+{\bar{R}}_{g}sin^{2}(s/2)] \end{array}$$

Given that the incident angle z, the slope angle s, and the incident angle of direct solar radiation θ are known (note: the calculation methods of these angles are described in Appendix A), theoretically, as long as the solar irradiance results of two or more directions are measured, $E_{dir}$ and $E_{sca}$ could be solved according to Equation (5).

Based on the above-mentioned principle, we developed a multi-angle light intensity sensor that can measure direct solar radiation and scattering radiation directly. The appearance is shown in Figure 1, i.e., five completely identical light intensity sensors are separately installed on the top, as well as the front, left, back, and right with a tilting angle of 15° of a transparent hemisphere (note: actually, exposing the photosensitive point will be better because it can avoid the reflection of the hemisphere surface; and the hemisphere is just for the convenience of controlling the tilting angle and the protection of the sensors), and these sensors are used to measure the light intensity facing different directions in real time so as to achieve the separation of direct and scattering radiation. For the solar spectrum in the visible to near-infrared range, the proportion of direct solar radiation at different wavelengths is roughly equal. So, the sensors sensitive to the visible and near-infrared spectral range can be selected for light intensity measurement. In this study, five GY-485-44009 light intensity sensors with a measurement range of 0–188,000 lux are used, and typically, the illuminance near the land surface cannot exceed the maximum. A USB-to-RS485 converter is used with a 5V DC power supply and data transmission (Baud rate: 9600 bps or 115,200 bps), and data collection is achieved by sending commands to each device address in sequence.

According to Equation (5), after determining the incident angle z, the slope angle s, and the incident angle of direct solar radiation θ, as well as the measured values of total solar irradiance from the above-mentioned five sensors, the direct and scattering radiation can be solved by using the overdetermined linear system described in Equation (6). Actually, Equation (6) can be written briefly and clearly in vector and matrix formats, and the expression is shown in Equation (7).(6)$$\left\{\begin{array}{l} E_{top}=E_{dir}[cosz_{top}+{\bar{R}}_{g}cos\theta_{top}sin^{2}(s_{top}/2)]+E_{sca}[cos^{2}(s_{top}/2)+{\bar{R}}_{g}sin^{2}(s_{top}/2)]\\ E_{front}=E_{dir}[cosz_{front}+{\bar{R}}_{g}cos\theta_{front}sin^{2}(s_{front}/2)]+E_{sca}[cos^{2}(s_{front}/2)+{\bar{R}}_{g}sin^{2}(s_{front}/2)]\\ E_{left}=E_{dir}[cosz_{left}+{\bar{R}}_{g}cos\theta_{left}sin^{2}(s_{left}/2)]+E_{sca}[cos^{2}(s_{left}/2)+{\bar{R}}_{g}sin^{2}(s_{left}/2)]\\ E_{back}=E_{dir}[cosz_{back}+{\bar{R}}_{g}cos\theta_{back}sin^{2}(s_{back}/2)]+E_{sca}[cos^{2}(s_{back}/2)+{\bar{R}}_{g}sin^{2}(s_{back}/2)]\\ E_{right}=E_{dir}[cosz_{right}+{\bar{R}}_{g}cos\theta_{right}sin^{2}(s_{right}/2)]+E_{sca}[cos^{2}(s_{right}/2)+{\bar{R}}_{g}sin^{2}(s_{right}/2)] \end{array}\right.$$(7)$$\underset{A}{\underbrace{\left[\begin{array}{ll} cosz_{top}+{\bar{R}}_{g}cos\theta_{top}sin^{2}(s_{top}/2) & cos^{2}(s_{top}/2)+{\bar{R}}_{g}sin^{2}(s_{top}/2)\\ cosz_{front}+{\bar{R}}_{g}cos\theta_{front}sin^{2}(s_{front}/2) & cos^{2}(s_{front}/2)+{\bar{R}}_{g}sin^{2}(s_{front}/2)\\ cosz_{left}+{\bar{R}}_{g}cos\theta_{left}sin^{2}(s_{left}/2) & cos^{2}(s_{left}/2)+{\bar{R}}_{g}sin^{2}(s_{left}/2)\\ cosz_{back}+{\bar{R}}_{g}cos\theta_{back}sin^{2}(s_{back}/2) & cos^{2}(s_{back}/2)+{\bar{R}}_{g}sin^{2}(s_{back}/2)\\ cosz_{right}+{\bar{R}}_{g}cos\theta_{right}sin^{2}(s_{right}/2) & cos^{2}(s_{right}/2)+{\bar{R}}_{g}sin^{2}(s_{right}/2) \end{array}\right]}}\underset{x}{\underbrace{\left[\begin{array}{l} E_{dir}\\ E_{sca} \end{array}\right]}}=\underset{b}{\underbrace{\left[\begin{array}{l} E_{top}\\ E_{front}\\ E_{left}\\ E_{back}\\ E_{right} \end{array}\right]}}$$

### 2.2. Land Surface Reflectance Conversion

Usually, the aircraft itself carries a light intensity sensor that can measure the solar irradiance of each band. However, the altitude of an aircraft in motion is not horizontal, but has a large pitching or rolling angle, which will result in significant impacts on the measured results of light intensity. To obtain accurate results, the directly measured results in the tilting status need to be corrected. The correction method is described below.

The calculation formula of surface reflectance is shown in Equation (8).(8)$$R_{\lambda }=E_{r}(\lambda )/E_{g}(\lambda )=\pi L(\lambda )/E_{g}(\lambda )$$
where $R_{\lambda }$ denotes the reflectance at a wavelength of λ, $E_{r}$ denotes the total reflective power within a hemisphere in unit area and unit wavelength ($W/m^{2}/nm$), and $E_{g}$ denotes the ground-received irradiance ($W/m^{2}/nm$). $E_{r}$ and $E_{g}$ of each band can be obtained by the imaging system and the irradiance sensors, respectively. L is the radiance ($W/m^{2}/Sr/nm$) reflected by the object and recorded by each pixel in each band.

Therefore, as long as the radiance L(λ) and ground-received irradiance $E_{g}(\lambda )$ of each band are obtained, the surface reflectance can be derived. The calculation methods for these two parameters are described as follows:(1)Method for calculation of radiance L(λ)

It is necessary to convert the DNs into the radiance for each pixel in each band first, and then the result can be converted into reflectance data. However, the DNs recorded in each original multispectral image can be affected by exposure time, ISO speed, aperture size, vignette effect, and some other factors. Therefore, it is necessary to convert the DNs of the original image into a unified standard, and the conversion method is shown in Equation (9).(9)$$DN^{\prime }=(DN-DN_{black})/N/G_{sensor}/T_{e}\times A\times V$$
where $DN^{\prime }$ is the corrected DN of each band; DN is the original DN recorded in each band; $DN_{black}$ is the black level (i.e., the DN value without any illumination) of each band; N is the maximum number of grayscale (e.g., for the DJI Phantom 4M in this study, the maximum number is 65,535, i.e., $2^{16}-1$); $G_{sensor}$ is the sensor gain of each band to correct the sensitivity difference due to ISO speed; $T_{e}$ is the exposure time of each band; A is the sensor gain adjustment factor; V is the vignette correction factor. The above parameters are all calibrated rigorously in the laboratory. The values of $DN_{black}$, $G_{sensor}$, $T_{e}$, and A are recorded in the XMP (Extensible Metadata Platform) metadata of each multispectral image, and the vignette correction factor of each pixel can be calculated by Equation (10).(10)$$V(x,y)=1+k_{1}r+k_{2}r^{2}+k_{3}r^{3}+k_{4}r^{4}+k_{5}r^{5}+k_{6}r^{6}$$
where $r=\sqrt{(x-x_{0}{)}^{2}+(y-y_{0}{)}^{2}}$; x and y are the column number and row number of each pixel; $x_{0}$ and $y_{0}$ are the column number and row number of image center; $k_{1}$, $k_{2}$, $k_{3}$, $k_{4}$, $k_{5}$, and $k_{6}$ are the vignette calibration coefficients, which are also recorded in the XMP metadata of each multispectral image.

There is a simple linear transformation relationship between the $DN^{\prime }$ and L(λ) of each band, which can be described as Equation (11).(11)$$L(\lambda )=G_{\lambda }\times DN^{\prime }(\lambda )+B_{\lambda }$$
where L(λ) is the radiance of each band, $G_{\lambda }$ is the gain value, $B_{\lambda }$ is the bias value, and $DN^{\prime }$ is the corrected digital number of each band. Note: The parameters may vary for different cameras, so a group of calibration boards with known reflectance can be used to obtain the values of gain and bias.(2)Method for calculation of $E_{g}(\lambda )$

If solar radiation is used as the radiation source, there is a direct proportional relationship between the illuminance in photometry (unit: lux) and the irradiance in radiometry (unit: $W/m^{2}$). Despite the different units of measurement, the resulting direct and scattering proportions are consistent. After implementing the component separation of $E_{dir}$ and $E_{sca}$, the direct radiation proportion p ($p=E_{dir}/E_{sun}$, and $E_{sun}=E_{dir}+E_{sca}$) can be obtained, and then the directly measured irradiance E can be converted into the ground-received irradiance $E_{g}$.

If $E_{dir}=pE_{sun}$ and $E_{sca}=(1-p)E_{sun}$ are input into Equation (5), one can obtain the relationship between E and $E_{sun}$, and the result is expressed by Equation (12).(12)$$E=[pcosz+(1-p)cos^{2}(s/2)+{\bar{R}}_{g}(pcos\theta +1-p)sin^{2}(s/2)]E_{sun}$$

Consequently, $E_{g}$ in Equation (4) can be written in the form of Equation (13).(13)$$\begin{array}{l} E_{g}=E_{dir}cos\theta +E_{sca}\\ \;\;\;\;\;=\;(pcos\theta +1-p)E_{sun}\\ \;\;\;\;\;=\frac{pcos\theta +1-p}{pcosz+(1-p)cos^{2}(s/2)+{\bar{R}}_{g}(pcos\theta +1-p)sin^{2}(s/2)}E \end{array}$$

Eventually, after obtaining the radiance L(λ) and the ground-received irradiance $E_{g}(\lambda )$, the corresponding surface reflectance $R_{\lambda }$ can be derived according to Equation (8).

## 3. Experiments

To verify the actual effect of the proposed reflectance conversion method in this study, a radiation component separation experiment and drone photogrammetry experiment were carried out simultaneously on 20 December 2024. The test region is located at the golf course of Linyi University. A DJI Phantom 4M multispectral drone (Dajiang Innovation Technology Co., Ltd., Shenzhen, China) was used as the remote sensing platform. A built-in stabilized imaging system with six cameras (1/2.9′’ CMOS sensor), involving an RGB (JPEG image) camera and five multispectral cameras including the blue (B), green (G), red (R), red-edge (RE), and near-infrared (NIR) bands (TIFF image), are equipped on the drone. Each camera has a spatial resolution of 1600 pixels × 1300 pixels, and it has a fixed aperture (the f-number is 2.2) and focal length (the focal length is 5.74 mm). The description of each multispectral band is shown in Table 1.

Additionally, an integrated solar irradiance sensor is mounted on the top of the aircraft. The sensor can capture synchronous solar irradiance information for each multispectral image, and the information is recorded in the XMP metadata of each image. Please refer to the document “https://dl.djicdn.com/downloads/p4-multispectral/20200717/P4_Multispectral_Image_Processing_Guide_EN.pdf (accessed on 20 December 2024)” provided by DJI for more detailed descriptions. The flight height is set to 45 m, and the image resolution is about 2.3 cm. The heading overlap rate is 80%, and the side overlap rate is 60%. There are 13 flight routes (as shown in Figure 2) that cover an area of about 200 m × 150 m (118°16′44.8″ E–118°16′53.5″ E, 35°06′52.9″ N–35°06′58.4″ N). If scattering radiation is dominated (e.g., on an overcast day), the light intensity sensor is less affected by the altitude changes of the aircraft. On the contrary, if direct radiation is dominated (e.g., on a very clear day), the scattering part can be almost ignored, and usually satisfactory results can also be obtained by using the cosine function correction to directly measure light intensity values. Moreover, if the flight direction is perpendicular to the sun’s direction, it can alleviate the impact of sensor tilt. In all cases, the worst scenario is that the light intensity fluctuates due to cloud cover, and the flight direction is consistent with the sun’s direction. In this case, the light intensity sensor is most sensitive to altitude changes, and the cosine function correction is also ineffective in achieving good results. To verify the effectiveness of the proposed method in this study, the test was implemented in the most challenging scenario, i.e., it was carried out on a cloudy day, and the flight direction was intentionally set to be approximately consistent with the sun’s direction. The total duration of the photogrammetry process is approximately 12 min (10:30:34–10:42:32). There are 338 captures involving the five multispectral bands. Note: The flight route planning application we used allows the imaging system to be in a shooting state even at turns, which leads to rapid altitude changes that result in inaccuracies of the drone-carried light intensity sensor. Hence, images captured at both ends of each flight route need to be discarded (30 captures are involved).

First, radiation component separation is carried out synchronously with the aerial photogrammetry using the self-developed multi-angle light intensity device, and the measured results are shown in Figure 3. Further, in order to obtain the gain value $G_{\lambda }$ and the bias value $B_{\lambda }$ using Equation (11), four calibration boards (see Figure 4) were used in this study. The reflectance of the calibration boards was measured by using a PSR-1100 hyper-spectrometer in advance, and the results are shown in Figure 5.

## 4. Results and Analysis

### 4.1. Radiation Components

The least square method is used to solve the overdetermined equation of Ax=b constructed by Equation (7). The measured results of direct and scattering radiation are presented in Figure 6. Consequently, the direct radiation proportions corresponding to the entire image capture time can be obtained, and the results are shown in Figure 7. Note: It has been tested that discarding the sensor-measured results with excessively large incident angles can enhance the accuracy and robustness of the results.

### 4.2. Reflectance Conversion

The results of $G_{\lambda }$ and $B_{\lambda }$ for each band were obtained using the calibration boards (the results are shown in Table 2), and then the radiance L(λ) was calculated by Equation (11).

The ground-received irradiance $E_{g}$ was calculated using Equation (13). By comparing the irradiance results directly recorded by the light intensity sensor mounted on the drone (see Figure 8a) with the corrected ground-received irradiance $E_{g}$ (see Figure 8b), the correction effect of the proposed method in this study is evident. As shown in Figure 8a, drastic fluctuations occur at the end points of flight routes due to flight direction changes. Fortunately, the corrected results in Figure 8b can significantly reduce drastic fluctuations. Since the top sensor remained horizontal throughout the testing process, its measured values should linearly correspond to the irradiance received by a horizontal ground surface. To facilitate comparison, the measured values of the top sensor in Figure 3 are divided by 100,000, and the results generate a reference line. Then, the reference line is added to Figure 8a and Figure 8b, and the results are shown in Figure 8c and Figure 8d, respectively. As shown in Figure 8d, the profiles of the corrected irradiance curves closely match the shape of the reference line, differing only by a certain scaling factor. Therefore, the results above validate the effectiveness of the proposed method in this study.

Additionally, from an efficiency perspective, the device we used takes about 0.27 s to collect the five light intensity data points, and the subsequent calculation takes only about 0.002 s. The time interval for image acquisition is approximately 2 s, so real-time correction of the light intensity values can be easily achieved.

Eventually, the drone-based multispectral remote sensing images were converted into land surface reflectance data using Equation (8). Moreover, the individual images above were mosaiced together into a complete reflectance image covering the entire test region after performing photogrammetric orthorectification, and the mosaiced result is shown in Figure 9.

### 4.3. Accuracy Validation

To validate the accuracy of the calculated reflectance results, some sample regions with a relatively large area and uniform spectrum were selected. Six typical land cover types (as shown in Figure 10), i.e., lake water, slab stone, shrub, green grass, red grass, and dry grass, were included, and their respective spectrum results were measured in situ by using the PSR-1100 hyper-spectrometer, synchronized with aerial photogrammetry, and the results are shown in Figure 11. Then, the land surface reflectance results calculated from the multispectral images were compared with the results measured in situ by using the PSR-1100 hyper-spectrometer (as shown in Table 3). It should be noted that lake water contains some impurity substances, and all other land covers are not absolute uniform reflectors either. Our approach involves measuring several samples for each land cover type and calculating the mean values.

Measurement errors are inevitable. Numerous factors can lead to the inconsistency between the land surface reflectance calculated from remote sensing images and the in-situ measured result, e.g., the measurement error of the light intensity sensor mounted on a drone, the measurement error of the pose sensor (gyroscope), the time synchronization error of different sensors, the calibration boards and natural objects not being strict diffuse reflectors, the resolution difference between the multispectral cameras and the hyper-spectrometer probe for the sample observation, and the measurement errors of the hyper-spectrometer itself (note: the hyper-spectrometer is susceptible to ambient illumination intensity fluctuations during outdoor operation, thereby inducing calibration errors). The analysis results in Table 3 indicate that the maximum MAE corresponds to the error of green grass in the red band, and the value is 3.68%; green grass has the highest MAE among the five multispectral bands, with a value of 1.59%; and for all land cover types, the highest MAE of 1.01% is found in the red band. Although there are some inevitable errors in the land surface reflectance calculated from the multispectral remote sensing images, the overall accuracy remains considerably high.

## 5. Conclusions

In this study, a new technique was proposed to separate the solar radiation components using a self-developed multi-angle light intensity device. The measured results were used to carry out land surface reflectance conversion for the synchronously collected multi-spectral images. The experimental results show that the proposed method can effectively correct the tilting effect of the drone-carried light intensity sensor, and thereby the multispectral images can be consequently converted into accurate reflectance data. Even under complicated illumination conditions due to the simultaneous changes in light intensity and components, the proposed method exhibits strong adaptability in obtaining accurate reflectance data. The reflectance data can be directly used as the input of various remote sensing models for quantitative analyses. Due to the similarity between the multispectral properties and the hyperspectral ones, the proposed method may also be suitable for drone-based hyperspectral data processing. Therefore, this study provides a solution to effectively solve the problem of accurate land surface reflectance conversion under complicated illumination conditions; moreover, it also provides a significant reference for the potential quantitative application of drone-based remote sensing data.

In future practical applications, the self-developed multi-angle light intensity device can be miniaturized and directly mounted on a drone. Thus, the radiation intensity and the solar radiation components can be obtained simultaneously by the drone-carried sensor, and the drone-based multi-spectral images can be converted into accurate land surface reflectance data in the post-processing process; or perhaps, the data processing algorithm can be integrated into the firmware of multispectral cameras, which is expected to enable real-time data processing and direct output of land surface reflectance data. Thus, the data processing efficiency would be greatly improved. Therefore, this study also provides valuable guidance for the design of future multispectral drones.

## Figures and Tables

**Figure 1 sensors-25-02604-f001:**
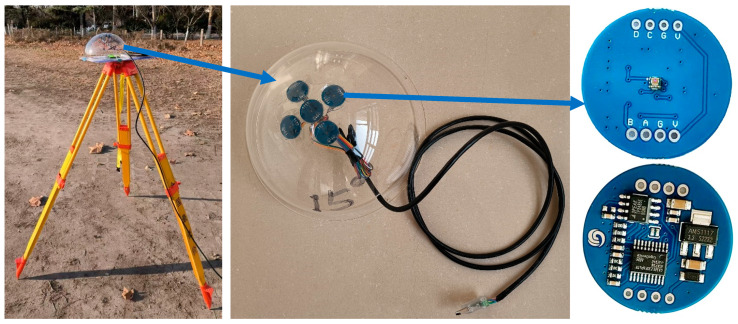
The self-developed solar radiation component separation device (note: the arrows indicate providing further details of the respective part).

**Figure 2 sensors-25-02604-f002:**
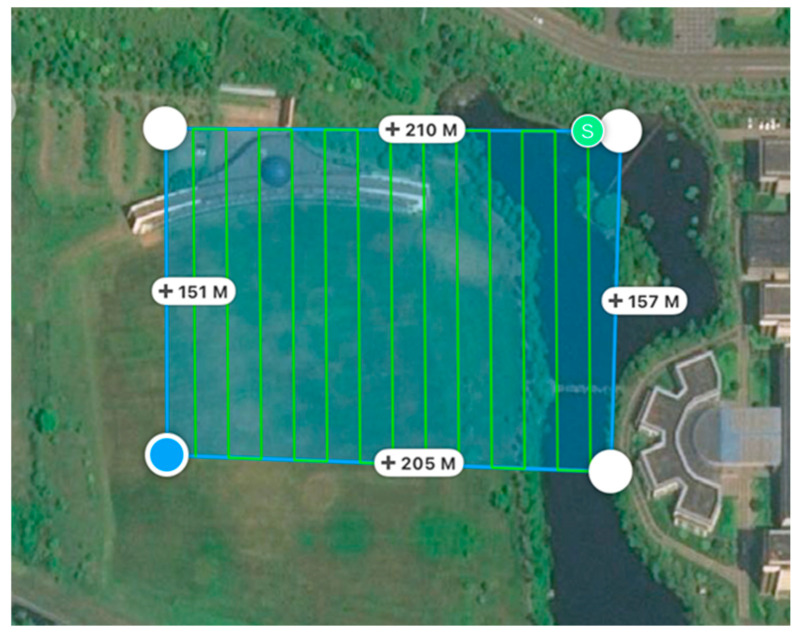
The flight routes in the test region (note: the green dot indicates the starting point of the route, and blue one indicate the end point).

**Figure 3 sensors-25-02604-f003:**
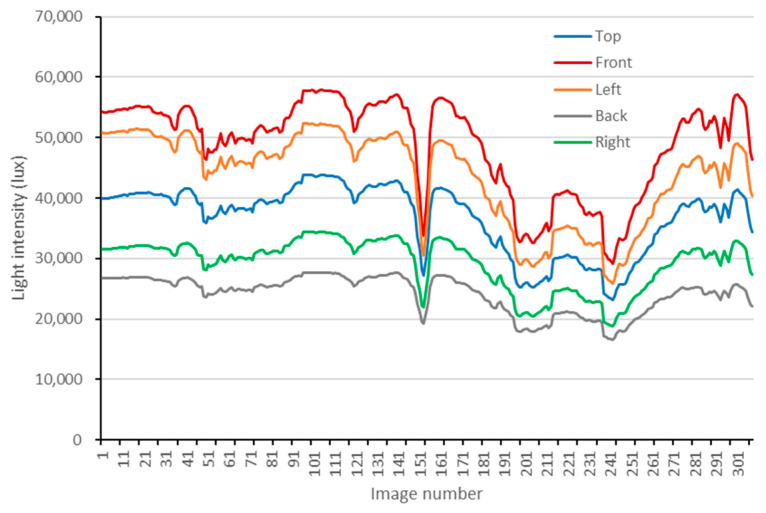
Light intensity measured by the self-developed multi-angle light intensity device in different directions corresponding to each image capture time (20 December 2024).

**Figure 4 sensors-25-02604-f004:**
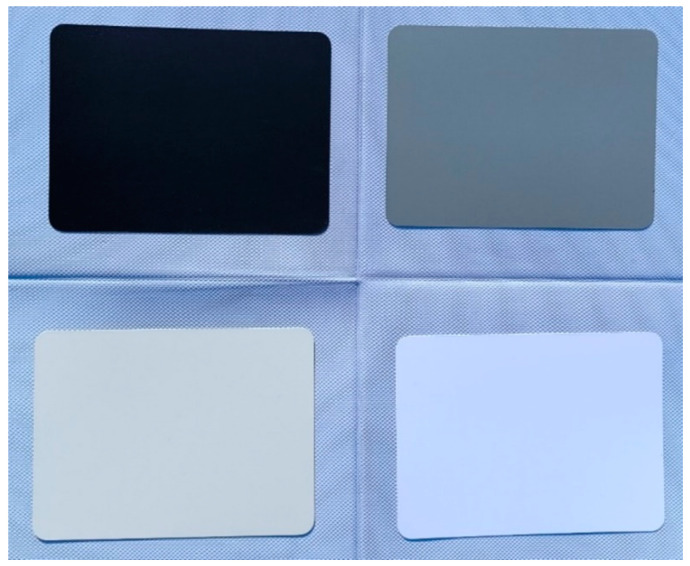
The calibration boards used in this study.

**Figure 5 sensors-25-02604-f005:**
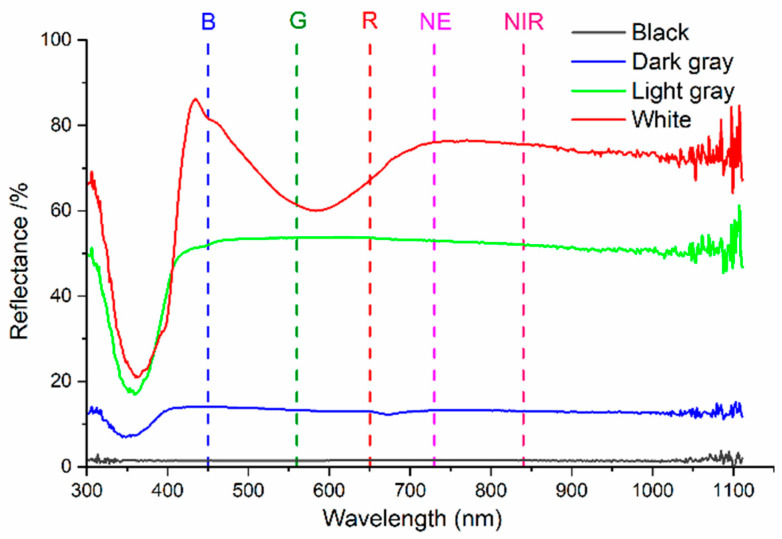
Reflectance of four calibration boards.

**Figure 6 sensors-25-02604-f006:**
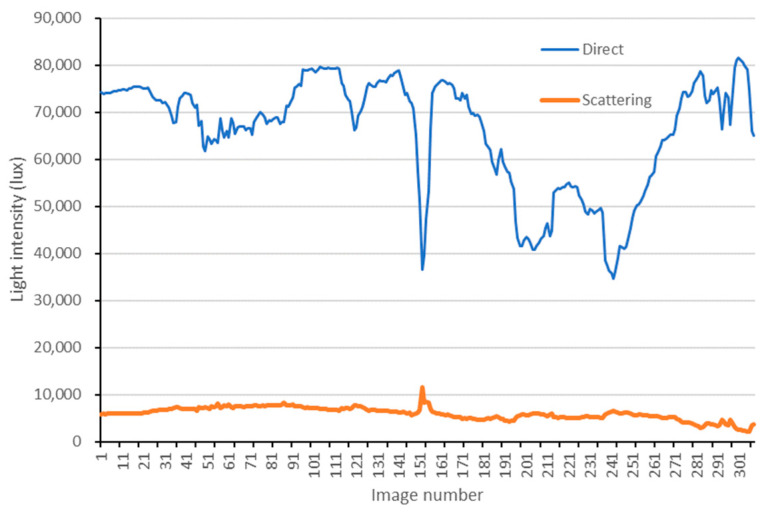
The results of direct and scattering radiation corresponding to the entire image capture time.

**Figure 7 sensors-25-02604-f007:**
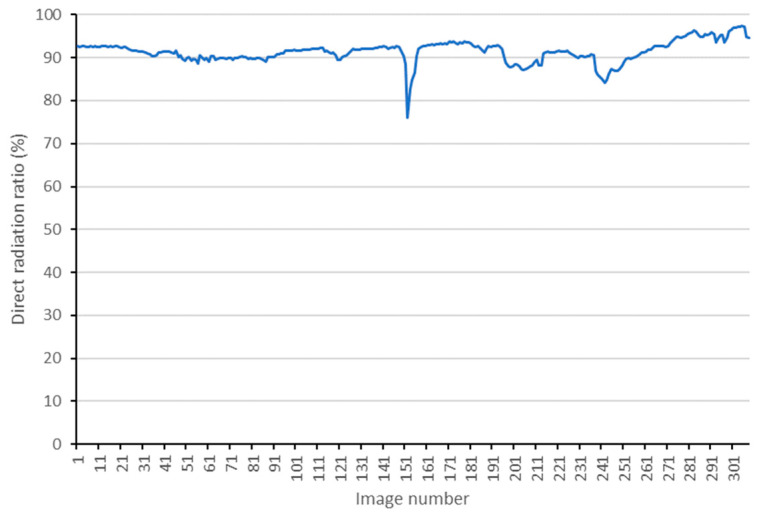
The results of direct radiation proportions corresponding to the entire image capture time.

**Figure 8 sensors-25-02604-f008:**
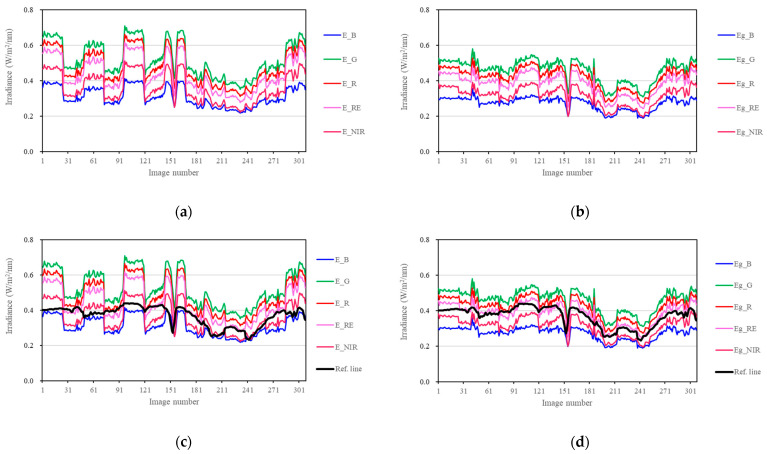
Irradiance results. (**a**) The results directly recorded by the light intensity sensor mounted on the drone; (**b**) the corrected ground-received irradiance $E_{g}$; (**c**) a reference line is added to (**a**); (**d**) a reference line is added to (**b**).

**Figure 9 sensors-25-02604-f009:**
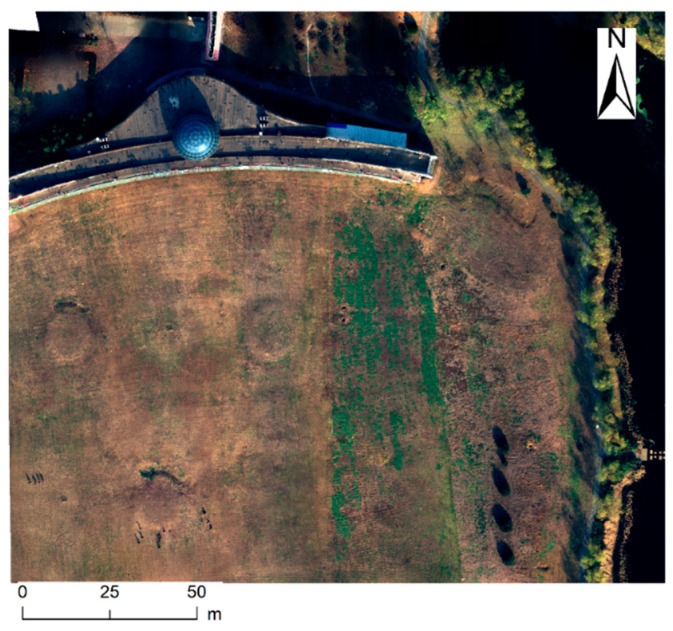
Mosaiced multispectral reflectance image after orthorectification (RGB = 3, 2, 1).

**Figure 10 sensors-25-02604-f010:**
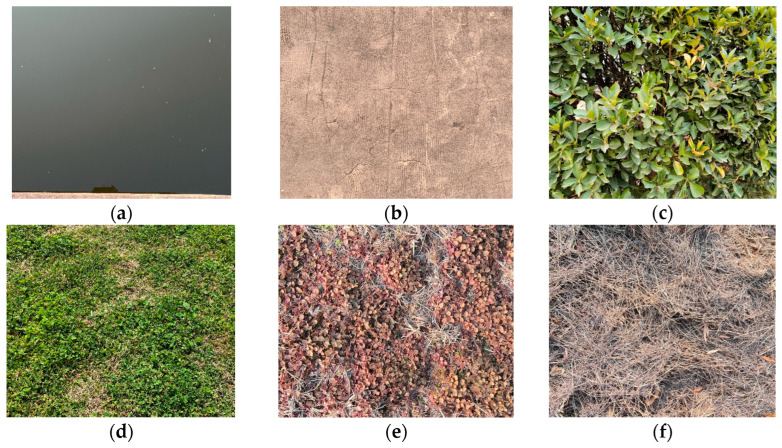
Samples of six typical land cover types. (**a**) Lake water; (**b**) slab stone; (**c**) shrub; (**d**) green grass; (**e**) red grass; (**f**) dry grass.

**Figure 11 sensors-25-02604-f011:**
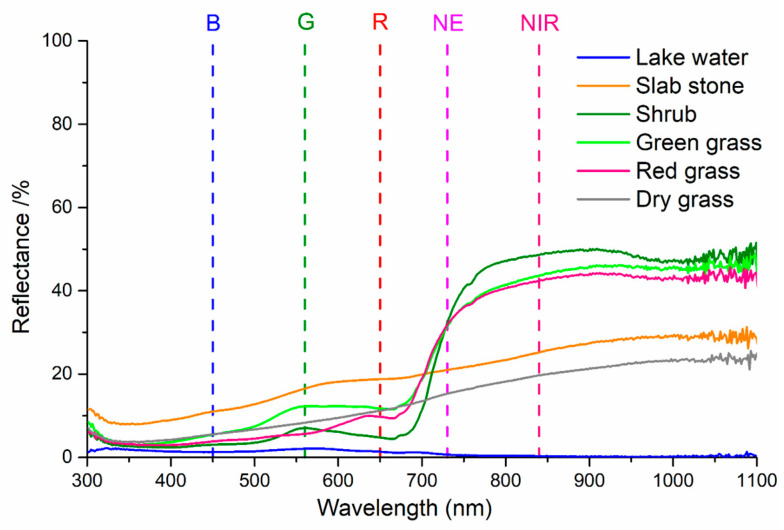
Spectrum results of six typical land cover types measured by PSR-1100 hyper-spectrometer.

**Table 1 sensors-25-02604-t001:** Parameters of DJI Phantom 4M multispectral camera.

Bands	B	G	R	RE	NIR
Wavelength range	450 nm ± 16 nm	560 nm ± 16 nm	650 nm ± 16 nm	730 nm ± 16 nm	840 nm ± 26 nm

**Table 2 sensors-25-02604-t002:** Gain and bias values of each band obtained by using by calibration boards.

Bands	B	G	R	RE	NIR
Gain ($G_{\lambda }$)	0.8375	0.6989	0.7742	0.9955	0.8153
Bias ($B_{\lambda }$)	−4.3830	−8.5053	−7.9623	−13.4527	−16.2095

**Table 3 sensors-25-02604-t003:** A comparison of the calculated land surface reflectance results from the multispectral images with the in situ measured results using the PSR-1100 hyper-spectrometer.

Land Cover Types		B (%)	G (%)	R (%)	RE (%)	NIR (%)	MAE (%)
Lake water	Calculated	1.77	2.08	1.59	1.46	1.38	
	Measured	1.29	2.06	1.34	0.69	0.31	
	Errors	0.48	0.02	0.25	0.77	1.07	0.52
Slab stone	Calculated	9.71	15.76	19.17	22.60	27.28	
	Measured	10.96	16.46	18.76	20.96	25.24	
	Errors	−1.24	−0.71	0.41	1.63	2.03	0.82
Shrub	Calculated	3.34	5.73	5.00	30.02	47.34	
	Measured	3.03	6.81	4.77	31.15	48.69	
	Errors	0.31	−1.08	0.24	−1.13	−1.35	
Green grass	Calculated	5.08	9.92	8.08	29.64	44.22	
	Measured	5.45	12.12	11.76	30.78	43.67	
	Errors	−0.36	−2.20	−3.68	−1.14	0.55	1.59
Red grass	Calculated	4.74	6.31	9.34	31.17	40.99	
	Measured	3.82	5.67	9.71	31.16	42.41	
	Errors	0.93	0.64	−0.37	0.01	−1.42	0.67
Dry grass	Calculated	7.41	11.56	15.97	21.81	27.44	
	Measured	7.22	11.63	14.86	21.01	26.64	
	Errors	0.19	−0.07	1.11	0.80	0.81	0.60
MAE (%)		0.59	0.79	1.01	0.91	1.21	

## Data Availability

The data are available from the corresponding author.

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
