# Peer review of "Accurate Conversion of Land Surface Reflectance for Drone-Based Multispectral Remote Sensing Images Using a Solar Radiation Component Separation Approach"

_sensors, 2025, doi:10.3390/s25082604_

Round 1
Reviewer 1 Report
Comments and Suggestions for Authors
This paper proposed a novel method to separate solar radiation components through a self-developed multi angle intensity device, thereby converting unmanned aerial vehicle multispectral images into high-precision surface reflectance data.
- There is a lack of systematic summary of the limitations of existing methods in the research background, especially insufficient discussion of the latest developments in radiation component separation technology.
- The derivation process of formulas (5) and (7) is relatively brief, which may affect the understanding of the readers for the key steps.
- The specific technical parameters of multi angle light intensity equipment, such as sensor model and calibration method, have not been fully described.
- It is suggested to add experimental data from different locations and environments to verify the universality of the method.
- The design of flying in the same direction as the incident direction of the sun may mask the limitations of the method. It is suggested to supplement comparative experiments with different flight directions (such as parallel to the direction of the sun's incidence) to analyze the impact of attitude angle changes on the results.
- The errors in the green and red bands of the green grass in Table 3 are relatively high, but the reasons have not been analyzed in depth.
- The display of irradiance correction effect in Fig. 8 is not intuitive enough. It is suggested to increase the statistical comparison between raw data and corrected data (such as standard deviation, correlation coefficient), and annotate the attitude angle changes at key time points.
- In the experiment, the authors should first provide relevant parameter and hardware settings to provide a foundation for subsequent experiments.
- It is suggested to discuss the potential challenges of future applications and methods in real-time processing, such as computational efficiency, in the paper.
Reviewer 2 Report
Comments and Suggestions for Authors
The authors described in detail a new method of accounting for scattered radiation in multispectral photography. Of course, this method is necessary to obtain more accurate remote sensing results. However, there are significant flaws in the article. The authors have not been able to prove the operability of the method.
- Table 3 shows the comparison data with measurements on a hyperspectrometer. But everything that the reader can understand from Table 3, the presented method gives data with a big error!
- To confirm the operability of the method, in Table 3 it is necessary to provide the survey data processed in the traditional way, if the author's method shows a smaller error, this will be proof of operability.
- The authors need to describe in more detail how the hyperspectrometer took pictures of what they claim to be homogeneous areas. What is the area of these plots? Where exactly were these sites selected in Fig. 9? Was the shooting carried out completely at the same time, or was there still a small time gap?
- There are especially many questions about measuring the water surface! Water is the only surface whose reflection coefficient can be accurately calculated. Its value is about 2%. However, with hyperspectral photography, it is often possible to distinguish reflection components from the near bottom of a reservoir in the spectral layers, so that the reflection coefficient becomes slightly higher than 2%. This raises additional questions about the accuracy of the hyperspectrometer survey. As can be seen from Table 3, only in the green spectrum the value of the reflection coefficient corresponds to the real situation.
Round 2
Reviewer 1 Report
Comments and Suggestions for Authors
I do not have any other questions.
Author Response
Comments 1: I do not have any other questions.
Response 1:Thank you for your affirmation of this paper.
Reviewer 2 Report
Comments and Suggestions for Authors
At a wavelength of 480 nm, a temperature of 20 °C, atmospheric pressure and a salinity of 35 °, the refractive index is 1.34509 (for pure water — 1.337). As you can see, the refractive index of even seawater does not change much, and at the same time it increases. That is, my remark regarding the incorrect spectrum of the lake surface remains valid. The authors need to either explain this correctly, or check the calibration of the hyperspectrometer. The more likely reason is still a hyperspectrometer calibration error.
Author Response
Comments 1: At a wavelength of 480 nm, a temperature of 20 °C, atmospheric pressure and a salinity of 35 °, the refractive index is 1.34509 (for pure water — 1.337). As you can see, the refractive index of even seawater does not change much, and at the same time it increases. That is, my remark regarding the incorrect spectrum of the lake surface remains valid. The authors need to either explain this correctly, or check the calibration of the hyperspectrometer. The more likely reason is still a hyperspectrometer calibration error.
Response 1: Thank you for your comments and suggestions, and I would like to express my respect for your rigor. Although measurement errors persist, all experimental measurements were retained as raw records. Measurement errors may also exist in other land cover types besides lake water, yet the underlying causes remain traceable. The calibration error of a hyperspectrometer is one of the main error sources. The reason for measurement errors was added in lines 432-433 (highlighted in green) -- (note: the hyper-spectrometer is susceptible to ambient illumination intensity fluctuations during outdoor operation, thereby inducing calibration errors).